

# Arctic observations of Hydroperoxymethyl Thioformate (HPMTF) – seasonal behavior and relationship to other oxidation products from Dimethyl Sulfide at the Zeppelin Observatory, Svalbard

Karolina Siegel[1,2,3], Yvette Gramlich[1,3], Sophie L. Haslett[1,3], Gabriel Freitas[1,3], Radovan Krejci[1,3], Paul Zieger[1,3] and Claudia Mohr[1,3]

[1]Department of Environmental Science, Stockholm University, Stockholm, SE-10691, Sweden
[2]Department of Meteorology, Stockholm University, Stockholm, SE-10691, Sweden
[3]Bolin Centre for Climate Research, Stockholm University, Stockholm, SE-10691, Sweden

*Correspondence to*: Claudia Mohr (claudia.mohr@aces.su.se)

**Abstract.**

Dimethyl sulfide (DMS), a gas produced by phytoplankton, is the largest source of atmospheric sulfur over marine areas. DMS undergoes oxidation in the atmosphere to form a range of oxidation products, out of which methanesulfonic acid (MSA) and sulfuric acid (SA) are well-known for participating in the formation and growth of atmospheric aerosol particles. Recently, a new oxidation product of DMS, hydroperoxymethyl thioformate (HPMTF) was discovered and later also measured in the atmosphere. Little is still known about the fate of this compound and its potential to partition to the particle phase. In this study, we present a full year (2020) of concurrent gas- and particle-phase observations of HPMTF, MSA, SA and other DMS oxidation products at the Zeppelin Observatory (Ny-Ålesund, Svalbard) located in the Arctic. This is the first time HPMTF has been measured in Svalbard and attempted to be observed in atmospheric particles. The results show that gas-phase HPMTF concentrations largely follow the same pattern as MSA during the sunlit months (April–September), indicating production of HPMTF around Svalbard. However, HPMTF was not observed in significant amounts in the particle phase, despite high gas-phase levels. Particulate MSA and SA were observed during the sunlit months, although the highest median levels of particulate SA were measured in February, coinciding with the highest gaseous SA levels with assumed anthropogenic origin. We further show that gas- and particle-phase MSA and SA are coupled in May–July, whereas HPMTF lies outside of this correlation due to the low particulate concentrations. These results provide more information about the relationship between HPMTF and other DMS oxidation products in a part of the world where these have not been explored yet, and about HPMTF's ability to contribute to particle growth and cloud formation.





## 1 Introduction

Oceanic dimethyl sulfide (DMS) is one of the largest contributors to atmospheric sulfur (17.6-34.4 Tg S yr$^{-1}$, Lana et al., 2011) and the most important source in marine areas. Up to 42% of global natural sulfur emissions can be traced back to DMS (Simó, 2001), which is equal to at least 50% of the total amount from anthropogenic sources (Klimont et al., 2013; Simó, 2001). DMS is a gas produced by algal communities when sea surface temperatures and sunlight conditions are favorable for primary production (Liss et al., 1993). When emitted to the atmosphere, DMS is oxidized to a range of gas-phase (g) sulfuric chemical species (Barnes et al., 2006; Yin et al., 1990). Some of these have a low enough volatility to condense to the particle phase (p), and have been shown to be able to participate in new particle formation (NPF, e.g. Beck et al., 2021; Lovejoy et al., 2004; Rosati et al., 2021) or contribute to the growth of aerosol particles. This has implications for the particles' ability to act as cloud condensation nuclei (CCN, see e.g. review by Ayers and Gillett, 2000), i.e. to form clouds in the atmosphere.

Clouds are important for the Earth's climate as they influence the radiation balance. In the Arctic, the common cloud type is low-level and mixed-phase (consisting of both liquid droplets and ice crystals) stratocumulus (*e.g.*, Shupe, 2011; Tjernström et al., 2012). Stratocumulus clouds are considered to be an important factor (Serreze & Barry, 2011; Wendisch et al., 2019) in the rapid increase of average surface temperature that has been observed in the Arctic region during the last decades (two to four times larger than the global average of +1°C compared to preindustrial times) (Rantanen et al., 2022). A more detailed understanding of aerosol and CCN chemistry, sources and seasonal variability in the Arctic is therefore needed to be able to make better predictions of future climate change (Schmale et al., 2021).

The oxidation scheme of DMS is highly complex, and represented in a poor or very limited manner in many atmospheric chemistry models. The scheme can however be described as mainly a two-route mechanism (Fig. 1): the addition pathway, where either a hydroxyl (OH), nitrate (NO$_3$) or halogen radical (e.g. bromine oxide, BrO, or chlorine, Cl) is added, or hydrogen abstraction pathway, in which a hydrogen (H) atom is removed. Main products in the abstraction pathway are the inorganic compound sulfuric acid (SA, H$_2$SO$_4$) (via sulfur dioxide, SO$_2$ and sulfur trioxide, SO$_3$), and the organic compound methanesulfonic acid (MSA, CH$_3$SO$_3$H). The first stable product in the addition pathway is dimethyl sulfoxide (DMSO, CH$_3$SOCH$_3$), followed by methanesulfinic acid (MSIA, CH$_3$S(O)OH) (Barnes et al., 2006). MSIA can either undergo reactive uptake to the particle phase or oxidize further to MSA and SA, although the abstraction pathway is normally considered more



important for the production of these two species. The different oxidation pathways further depend on the ambient temperature, where the abstraction mechanism has been shown to be favored at higher temperatures and the addition mechanism at lower temperatures (Hoffmann et al., 2021; Wollesen de Jonge et al., 2021). It is therefore plausible that the addition pathway is of
higher importance in the Arctic, where temperatures are generally low.

Recently, a new oxidation product of DMS was discovered in situ during an aircraft campaign by Veres et al. (2020), hydroperoxymethyl thioformate (HPMTF, $C_2H_3OSO_2H$) (see Fig. 1). This compound is produced via the H-abstraction pathway, leading to the methylthiomethyl-peroxy radical ($CH_3SCH_2OO^\bullet$), quickly followed by an H-shift to form intermediate radicals and then HPMTF (Berndt et al., 2019; R. Wu et al., 2015; Z. Wu et al., 2022). It was found to be ubiquitous (frequently
> 50 ppt) in the lowest kilometers of the troposphere in both spring (April–May) and autumn (September–October), and was detected up to 14 km altitude over large parts of the world's oceans between 80°N and 85°S latitudes. It was further shown to be readily removed through cloud uptake and there was no apparent relationship between HMPTF and MSA in the gas phase. The study also speculated that HPMTF could contribute to NPF or particle growth.

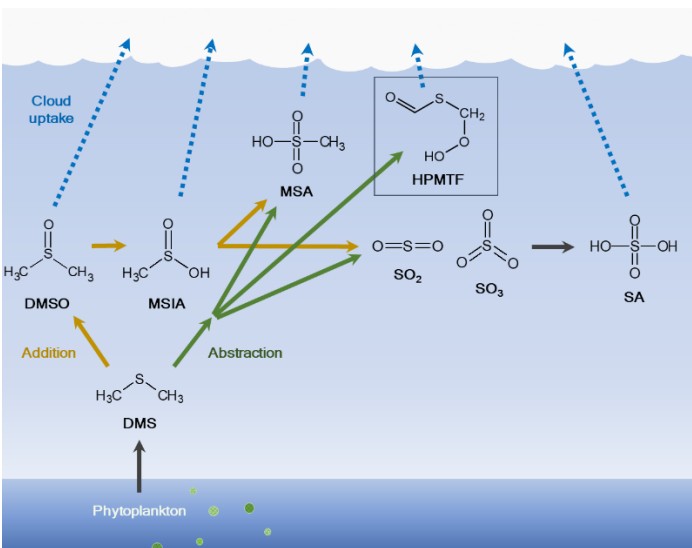

**Figure 1.** Simplified oxidation scheme of dimethyl sulfide (DMS) in the atmosphere. DMS is produced by microbiological activity in the ocean and emitted to the atmosphere, where it is oxidized through two main routes: 1) *addition* of a radical to produce dimethyl sulfoxide (DMSO) and methanesulfinic acid (MSIA), and further methanesulfonic acid (MSA) and/or sulfuric acid (SA) via sulfur dioxide (SO$_2$) and



sulfur trioxide (SO₃); 2) *abstraction* of a hydrogen (H) atom to produce MSA, hydroperoxymethyl thioformate (HPMTF; marked with a box) and/or SA via SO₂ and SO₃. The figure was created using information from Barnes et al. (2006) and R. Wu et al. (2015).

Since the discovery of HPMTF, it has inspired several new studies (e.g., Khan et al., 2021; Novak et al., 2021; Vermeuel et al., 2020; Wollesen de Jonge et al., 2021).

Fung et al., (2022) developed an understanding of where certain DMS oxidation products can dominate in the world due to variation in dominating DMS oxidation mechanisms. This clarified why the highest HPMTF concentrations were found over tropical oceans with a high degree of OH oxidation through the abstraction pathway, and lower over equally
microbiologically productive waters with a relatively higher degree of oxidation through the addition pathway, such as the North Atlantic and Canadian Arctic (Fung et al., 2022; Veres et al., 2020).

Wollesen de Jonge et al. (2021) showed based on a chamber study that HPMTF accumulates in the gas phase during cloud-free conditions, but does not significantly partition to the particle phase and is unlikely to contribute to NPF. However, it was speculated in this study that HPMTF could indirectly contribute to growth of aerosol particles through increased
aqueous-phase production of sulfate (SO₄²⁻). This does not necessarily happen where DMS concentrations are locally high, as HPMTF might be transported large distances before taking part in these partitioning and oxidation processes, depending on OH concentrations, cloud occurrence and ambient temperatures (Khan et al., 2021; Novak et al., 2021; Vermeuel et al., 2020).

In the Arctic, where aerosol-climate interactions are especially poorly understood (Schmale et al., 2021) and DMS emissions are expected to increase in the near future due to higher sea temperatures (Land et al., 2014), more measurements
of HPMTF during different seasons and atmospheric conditions could help to improve our knowledge of DMS oxidation and aerosol formation in this region. Hence, in this study we present observations of atmospheric DMS oxidation products, including HPMTF, from the Zeppelin Observatory in Ny-Ålesund, Svalbard (79 °N) throughout the year of 2020. The measurements were made using a high-resolution time-of-flight chemical ionization mass spectrometer with a filter inlet for gases and aerosols (FIGAERO-CIMS, Lopez-Hilfiker *et al.*, 2014) and iodide as the reagent ion. With the FIGAERO-CIMS,
gas- and particle-phase measurements can be made concurrently, which provides a unique possibility to study sources and phase-transition mechanisms at time resolutions of seconds to hours. In this project, we were able to study this over all seasons.





To our knowledge, this is the first time ambient HPMTF has been measured in Svalbard and the first time HPMTF detection

in the particle phase has been attempted in the ambient atmosphere.

## 2 Methods

### 2.1 The measurement site

The results presented in this study were derived from measurements made during the Ny-Ålesund Aerosol Cloud Experiment

(NASCENT) campaign in 2019–2020. The data presented in this study was collected between 2020-01-01 and 2020-12-17

(where entire August is missing due to malfunctioning instrumentation) at the Zeppelin observatory atop Mt. Zeppelin (474 m

a.s.l), 2 km south of Ny-Ålesund, Svalbard (78.9° N, 11.9° E) (Platt et al., 2022), see Fig. 2. A detailed description of the entire

campaign and overview on first results can be found in Pasquier et al. (2022). Here follows a description of the instrumentation

and methods used for this particular study.

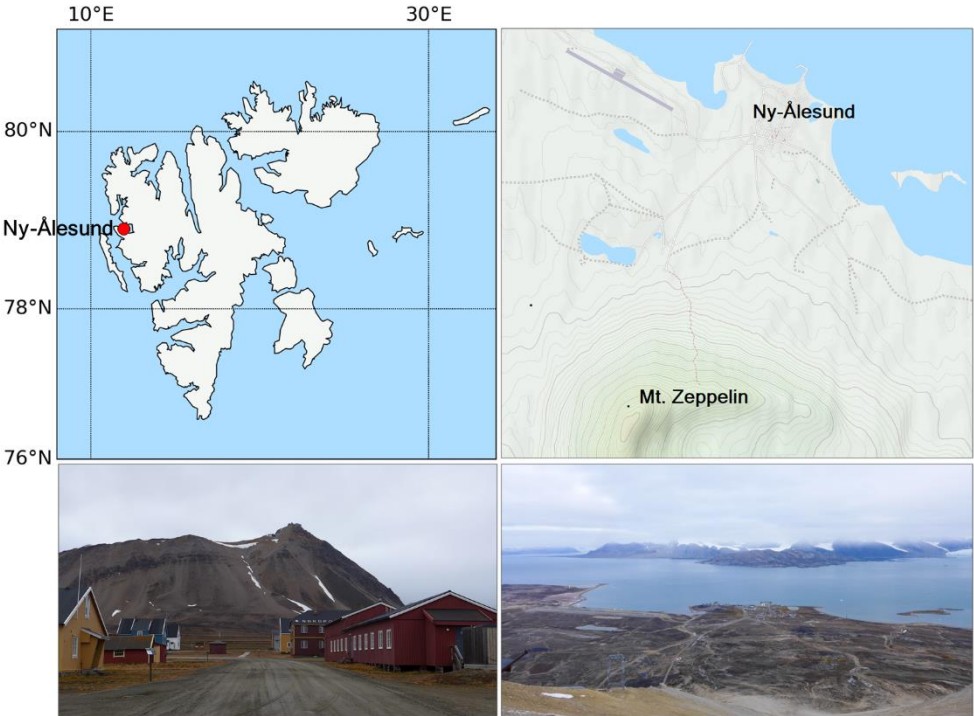

**Figure 2.** Upper left: map of Svalbard, where Ny-Ålesund is marked with a red circle. Upper right: Mt. Zeppelin with the Zeppelin

Observatory in relation to Ny-Ålesund (maps generated using Python's Matplotlib Basemap toolkit and ©OpenStreetMap contributors 2023,



distributed under the Open Data Commons Open Database License (ODbL) v1.0). Lower left: view of Mt. Zeppelin and the observatory

from Ny-Ålesund. Lower right: view of Ny-Ålesund and Kongsfjorden from the Zeppelin Observatory (photos taken in September 2021).

## 2.2 Chemical composition of gaseous and particulate atmospheric compounds

The chemical composition of atmospheric compounds in the gas-, particle-, and cloud phase was determined at molecular level

with a FIGAERO-CIMS (Aerodyne Research Inc., USA), with iodide (I$^-$) as the reagent ion. This setup allows for detection

of mainly polar and oxygenated organic compounds, due to the clustering mechanism of I$^-$ (Lee et al., 2014). The FIGAERO-

CIMS recorded data every second (1 Hz) until mid-February, and every two seconds (0.5 Hz) during the rest of the year. The

gas phase was sampled with a ¼ inch outer diameter (O.D.) and ~5 m long polytetrafluoroethylene (PTFE) tubing sticking out

~50 cm through a hole in the northern wall of the Zeppelin Observatory and with the tip protected by an attached funnel. The

aerosol particles were sampled through a gently heated whole-air inlet (which samples all aerosol and cloud particles smaller

than approx. 40 µm) at ~480 m a.s.l.. The aerosol inlet was connected to the FIGAERO-CIMS by stainless steel tubing (O.D.

½ inch, length: ~6 m) through a three-way switching valve. During cloudy conditions, defined as when the visibility at the

observatory was ≤ 1000 m, the three-way valve switched to sampling cloud particles (> 6–7 µm in diameter) through a ground

based counterflow virtual impactor (GCVI; Brechtel Manufacturing Inc., USA, Model 1205) inlet (Karlsson et al., 2021). After

drying of the cloud particles, the chemical composition of the remaining cloud residuals was measured with FIGAERO-CIMS.

The first results of this analysis are reported in Gramlich et al. (2022).

The FIGAERO inlet is designed in a way that enables alternating gas- and particle-phase measurements (Lopez-

Hilfiker et al., 2014; Thornton et al., 2020). Particles are deposited on a PTFE filter for a specified period of time, during which

gaseous compounds are sampled via the gas-phase inlet. When enough particulate matter has been collected onto the filter, the

gas-phase inlet is blocked and the filter moved into place in front of the instrument's inlet to enable particle-phase analysis.

This mechanism makes it possible to compare the two phases at the same point in time.

During NASCENT, a FIGAERO-CIMS measurement cycle was 210 min (Fig. A1 and Table A1), where the first 150

min were made up by gas-phase measurements: 3x40 min of ambient atmosphere measurements and 2x15 min of background

measurements with ultra-pure air ("zero air") from a generator (Teledyne API, USA, Model 701H), and simultaneous particle

deposition on the PTFE filter. The last 60 min of the measurement cycle consisted of particle desorption, where the compounds

130 collected on the filter were evaporated by a stream of ultrapure nitrogen ($N_2$) from a generator (Peak Scientific, UK, Model NG5000), gradually heated during 20 min from room temperature to ~200°C. The temperature was then kept at this level for another 20 min, allowing all compounds to evaporate completely off the filter and their corresponding signal return to background levels. This means that compounds that are not volatile enough to evaporate at ≤ 200°C (e.g. sea salt) (Rasmussen et al., 2017) or that decompose at these temperatures are not directly measurable by the FIGAERO-CIMS. Particle-phase

135 blanks (background signal) were collected every third cycle by sampling through another particle filter, which removed particles from the sampling flow upstream of the sample filter.

  The desorbed compounds were ionized by $I^-$ and separated by their time-of-flight (ToF) in the mass spectrometer. The mass-to-charge ratio (m/z) of each compound was determined from the ToF by using known ions as calibrants, from which their molecular compositions were determined.

140 **2.2.1 Data processing**

The FIGAERO-CIMS raw data were pre-averaged to 30 s time resolution. After mass calibration, the signals were normalized to the sum of the reagent ion signal ($I^-$, m/z: 126.905) and the cluster of iodide and water ($H_2OI^-$, m/z: 144.916). Inclusion of $H_2OI^-$ in the normalization accounts for changes in the water vapor content of the ambient air, and the humidity difference between the ambient gas-phase measurement compared to the particle desorption using dry $N_2$, which influences sensitivity of

145 iodide adduct ionization (Lee et al., 2014).

  The time period between two particle heatings was identified as the "sampling period", during which the gas-phase measurements took place (see Fig. A1 and Table A1). The sampling period was usually interrupted twice by two background measurements of 15 min each. The 30 first data points (~15 min) of the gas-phase signal were removed from each sample segment, since the signal took some time to stabilize after each shift (as is visible from the $HNO_3I^-$ signal in Fig. A1). Similarly,

150 to assure no inclusion of background signal in the averaged data of the sample, the 15 last data points (~7.5 min) were removed as well. Around 18 min of gas-phase data were used for analysis from each sample segment (represented by the green circles in Fig. A1). The segments were averaged individually, and the segment averages within the same sampling period were in turn averaged to a single data point for that period (dark-green triangles in Fig. A1). The average background signal (orange triangles in Fig. A1) of each sampling period was calculated in the same way, where only three data points (1.5 min) at the

end of each background period were used (yellow diamonds in Fig. A1). The background signal was thereafter subtracted from

the sample signal to give two sets of timeseries: 1) a background-subtracted average of the sampling period (using the values

represented by the dark-green and orange triangles in Fig. A1), which was used for comparison to the particle-phase signal of

the same period; 2) background-subtracted data with the full time-resolution of 30 s, to use when not comparing to particle-

phase data. The FIGAERO-CIMS was not calibrated to any known substance, and the unit of the gas-phase data is [number of

detected ions s$^{-1}$], i.e. not atmospheric concentration. The data can be used for qualitative analysis and for relative

quantification between measured gaseous compounds. To achieve atmospheric concentrations, calibration of the FIGAERO-

CIMS with known gaseous compounds is recommended for future studies.

The particle-phase signal was calculated as the signal integrated across time from the start of the heating (when

the temperature starts ramping up) until the end of the desorption time (when the temperature starts decreasing again) (see the

red line in Fig. A1). The signal was thereafter normalized to the sampled volume, as the sampling time could sometimes differ

between samples. Particle backgrounds were calculated as the interpolated time-integrated signal between two consecutive

particle blanks, which was in turn subtracted from the time-integrated particle sample signal. Handling blanks were not needed

as the FIGAERO operated automatically. However, the first particle sample after the filter had been exchanged was excluded

from the analysis due to the risk of contamination. Since the particle-phase data was time-integrated and normalized to the

sampled volume, the unit is therefore [number of detected ions per liter of sampled air], and the data can be used in the same

way as the gas-phase data (qualitative analysis and for relative quantification between measured particulate compounds). More

details on the processing of the particle-phase data is given by Gramlich et al. (2022).

For the results presented in Sect. 3.1, 3.2 and 3.4, we only consider cloud-free conditions and visibilities > 5

km (measured by the GCVI), since the presence of cloud water and ice could influence the chemical composition of gaseous

and particulate compounds. The data not considered in these sections are however included in the results of Sect. 3.3, where

the effect of visibility and relative humidity (RH) is investigated.

### 2.2.2 Peak resolution and separation

The mass resolving power of the ToF mass spectrometer used for this study was ~5000 m/Δm, which does not allow for full peak separation of the ions contributing to nominal m/z 235. The closest neighbor of HPMTF (m/z 234.893) is nitrogen pentoxide ($N_2O_5I^-$, m/z 234.885). $N_2O_5$ and HPMTF luckily rarely coexist in the lower troposphere, as HPMTF is produced during the sunlit months when $N_2O_5$ production is suppressed and vice versa (Veres et al., 2020). Although $N_2O_5$ is a highly reactive gaseous compound, it can partition to the particle phase but then largely dissociates (Gržinić et al., 2017). The second-closest neighbor to HPMTF is a cluster of acetic acid with an iodate ion ($IO_3^-CH_3COOH$, m/z 234.910) (Veres et al., 2020). Calculations of the statistical precision ($\sigma_B$) of the peak intensities (Cubison & Jimenez, 2015) of these two overlapping peaks for four example cases are presented in Fig. B1 and Table B1: (a) HPMTF as single ion (in practice meaning a high HPMTF/$IO_3^-H_3COOH$ ratio); (b) HPMTF as the larger (parent) peak and $IO_3^-CH_3COOH$ as the smaller (child) peak; (c) equal intensities of HPMTF and $IO_3^-CH_3COOH$; (d) $IO_3^-CH_3COOH$ as the parent peak and HPMTF as the child peak. As expected, $\sigma_B$ is the highest in (a) (51%) and the lowest in (d) (2.9%). However, it should be noted that the peak intensities of HPMTF and $IO_3^-CH_3COOH$ (p) seemingly are much too low (< 10 ions $s^{-1}$) to even be covered by the analysis in Cubison & Jimenez (2015).

### 2.3 Other particle measurements

A condensation particle counter (CPC; TSI Inc., USA, Model 3772) behind the whole-air inlet was used to measure the total particle number concentration (1 Hz data averaged to 1 min time resolution). An optical particle size spectrometer (FIDAS 200S, Palas GmbH, Germany) was used to measure particulate mass concentrations (PM1, PM2.5, PM4 and PM10 averaged to 1 h time resolution). The instrument was installed on the terrace of Zeppelin Observatory and was equipped with its own heated inlet and additional RH and temperature (T) sensors to ensure that all values were measured at dry conditions.

### 2.4 Trajectories and meteorological data

Backward trajectories (10 days, out of which 5 days were used for this study) were calculated with HYSPLIT (Stein et al., 2015) using 3-hourly archive data from the National Center for Environmental Prediction's (NCEP) Global Data Assimilation System (GDAS) with 1 degree horizontal grid resolution, starting from the Zeppelin Observatory at 474 m a.s.l.. Since gas-



and aerosol emissions from the surface (ocean) would be considered for the analysis, data within the model mixed-layer height were distinguished from the data points above the mixed layer. Monthly chlorophyll-*a* data from the Aqua/MODIS satellite were retrieved from NASA Earth Observatory (Hu et al., 2012).

Meteorological data (air temperature, atmospheric pressure and relative humidity) with 1 h time resolution were
downloaded from the EBAS data base (Norwegian Institute for Air Research). More meteorological details are presented in Pasquier et al., (2022).

## 3 Results and discussion

### 3.1 Seasonal pattern of DMS oxidation products

The high-resolution FIGAERO-CIMS data revealed the presence of seven compounds uniquely or potentially related to DMS
oxidation: (HPMTF: $C_2H_3OSO_2H$, MSA: $CH_3SO_3H$, SA: $H_2SO_4$, sulfur dioxide: $SO_2$, sulfur trioxide: $SO_3$, bisulfate: $HSO_3$, and disulfuric acid: $H_2O_7S_2$). This is largely similar to earlier findings of particle-phase DMS oxidation products in the High Arctic with FIGAERO-CIMS (Siegel et al., 2021). The two organic compounds MSA and HPMTF have no other precursors than DMS, whereas the five inorganic compounds (SA, $SO_2$, $SO_3$, $HSO_3$, and $H_2O_7S_2$) can originate from DMS oxidation but also from other sources, such as anthropogenic (e.g., ship fuel emissions) or other non-marine natural $SO_2$ sources (e.g.,
volcanic activity) (Barnes et al., 2006; Berndt et al., 2019; R. Wu et al., 2015). Here, we present the results of the gas- and particle-phase measurements of the DMS-related compounds.

### 3.1.1 Gas-phase DMS oxidation products

Phytoplankton need solar radiation to be able to produce DMS. Due to the large seasonal changes in the Arctic with the 24 h of darkness in winter and 24 h of sunlight in summer, it is of interest to compare the occurrence of DMS oxidation products
throughout a full year.

In Figure 3a, c and e we show boxplots (monthly medians and percentiles) of the temporal evolution of the inorganic compound SA and the organic HPMTF and MSA and between 2020-01-01 and 2020-12-17. A full time series, also including $SO_2$, $SO_3$, $HSO_3$ and $H_2O_7S_2$, can be found in Fig. C1.



The inorganic SA (g) (Fig. 3a), which could be linked both to DMS oxidation and anthropogenic sources, displays

the highest levels in January–April, i.e. not during the Arctic bloom period. During this time, synoptic-scale meteorology is

different than in summer, with air masses from lower latitudes being advected into the north polar region (e.g., Heintzenberg,

Hansson and Lannefors, 1981; Frossard et al., 2011). These air masses carry anthropogenic pollution with elevated

concentrations of sulfuric and other chemical species from e.g. industries, and are known as the Arctic haze phenomenon

(Hansen & Rosen, 1984; Mitchell, 1957). The results in Fig. 3a indicate that wintertime anthropogenic sources are the main

contributor of annual inorganic sulfur in Ny-Ålesund, in agreement to previous studies (e.g., Jang et al., 2021).

MSA (g) (Fig. 3b) follows the expected pattern for a compound with DMS as its only source, where the

concentration is very low during the winter and beginning of spring (October–March) and then starts increasing in April, when

DMS starts being produced in the vicinity of Svalbard (Jang et al., 2021). It should however be noted that the onset and

variability of the spring blooms show large variability from year to year (Galí et al., 2019; Jang et al., 2021). MSA (g) peaks

sometime in May–June, and thereafter decreases toward the end of summer (September).

The levels of HPMTF (g) (Fig. 3c) develop similarly to MSA to some degree, with low levels in March, an increase

in April to a peak in May. However, the variability over the bloom period is larger than for MSA (see Fig. C1), and the

measured HPMTF (g) concentration in May–September is sometimes as low as before the DMS production onset. HPMTF (g)

further displays elevated levels in January and throughout September–October, which are completely lacking for MSA (g).

During January and end of October, the Arctic experiences very few to no sunlit hours and DMS is not produced in the

immediate vicinity of Svalbard, meaning that the measured HPMTF must have been transported from an area with DMS

production during these months. Another possibility is that we instead of HPMTF (g) measured $N_2O_5$ (g) in winter. This is

further discussed in Sect. 3.3 and 3.4.

During April–June, when MSA and HPMTF (g) evolve in a common way, the level of MSA is around 5 times higher

than HPMTF (assuming they have the same sensitivity in the FIGAERO-CIMS). This ratio is similar to the simulation

**AtmMain** presented in Wollesen de Jonge et al. (2021), which was used as a base run scenario for an air parcel moving along

a trajectory in a marine environment. The run was in total 120 h and included eight in-cloud events during both day- and night-

time, with the last in-cloud event also including a rain event. The temperature was 280 K (~7°C) and the RH = 90%, i.e. fairly



similar to the conditions in summer during the NASCENT campaign (Fig. C2). Therefore, it can be assumed that this scenario

is representative of our data during this time of the year.

Further, the fact that HPMTF and MSA appear to have a relatively strong interrelationship in April–June indicates that they were formed from a local DMS source during this period (e.g. Becagli et al., 2016). Hence, we can assume that DMS was at least partly oxidized through the abstraction pathway (Fig. 1) during these months (Berndt et al., 2019; R. Wu et al., 2015), despite the low occurrence of this mechanism compared to the addition pathway in the Nordic Seas according to Fung

et al. (2022). However, the discrepancies that exist in the relative temporal evolution between measured MSA (g) and HPMTF (g) in summer could also be a sign that HPMTF is not produced as efficiently as MSA close to Svalbard due to e.g. low occurrence of OH radicals and/or meteorological factors. This hypothesis is supported by the higher importance of OH-initiated DMS oxidation at lower latitudes in the study by Fung et al. (2022), and the DMS conversion yields of MSA and HPMTF in the different model runs by Wollesen de Jonge et al. (2021), where the HPMTF/MSA ratio was considerably lower

in the **AtmMain** case compared to cases with higher temperature and lower RH. This could be both due to differing sources and sinks, such as dry/wet deposition of MSA (Bergin et al., 1995) and HPMTF (Khan et al., 2021), heterogeneous oxidation of MSA (g) (Mungall et al., 2018) and evaporation of MSA from the particle- to the gas phase (Baccarini et al., 2021).



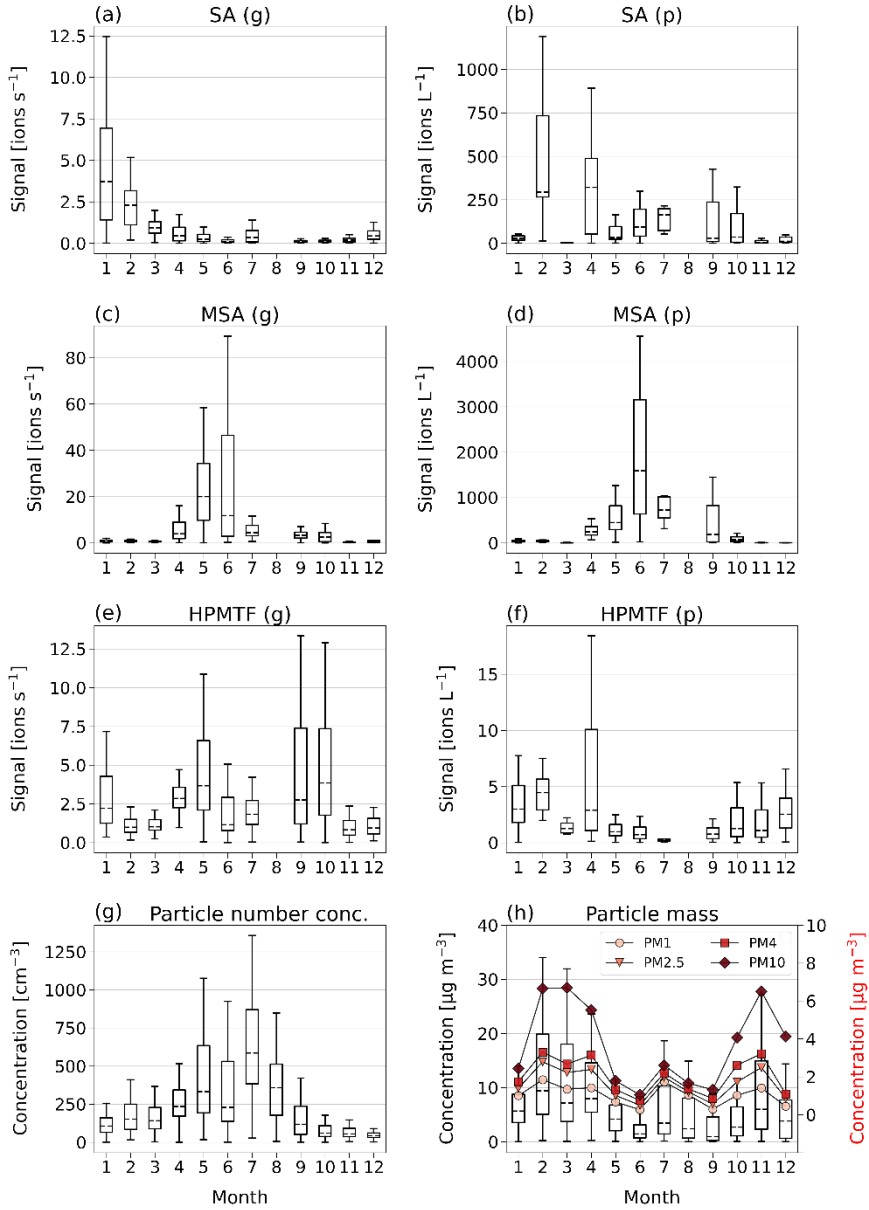

**Figure 3.** Box-and-whisker plots (with outliers removed) of SA, MSA and HPMTF in the gas- (g) and particle (p) phase per month of 2020,
analyzed by FIGAERO-CIMS (panel (a)-(f)). Dashed horizontal lines inside the boxes represent median values, boxes the 25th and 75th data
percentiles, and whiskers the maximum and minimum values of the populations. Panel (g) shows the total particle number concentration.
Panel (h) shows the total particle mass (PM) of particles with diameters 1, 2.5, 4 and 10 µm, where the box-and-whisker plots (left-side axis)
correspond to the sum of the particle sizes and the red markers (right-side axis) represent the monthly means of each size.



### 3.1.2 Particle-phase DMS oxidation products

Monthly boxplots of particle-phase SA, MSA and HPMTF are shown Figure 3b, d and f.

For SA (p) (Fig. 3b), the highest concentrations were measured in February and April, i.e. during the haze period. Slightly elevated levels are also found during summer, likely connected to DMS oxidation. These results are largely supported by measurements of particulate sulfate using an aerosol chemical speciation monitor (ACSM; data not shown), where the concentrations are overall declining from the beginning to the end of the year, with a small increase in July–August.

MSA (p) (Fig. 3d) follows a similar seasonal pattern in the particle- to the gas phase, where the highest levels are seen in June. The concentrations of HPMTF (p) (Fig. 3f) are much lower (2–3 orders of magnitude) than those of particulate MSA and SA and cannot be properly separated from the background. Hence, our conclusion is that the particulate amounts of HPMTF are negligible, and/or difficult to observe with the settings of our instrument. One reason for this is an extensive overlap with the peak $IO_3^-$ ($CH_3COOH$) (a cluster of acetic acid with an iodate ion, Veres et al., 2020), see Sect. 2.2.2. The

signal of this cluster increased in the particle phase during the summer months relative to the winter, causing the time series of HPMTF (p) to appear very different from what is expected, with the lowest levels in summer and the highest in winter and spring. The statistical precision ($\sigma_B$) of the peak intensities of these two ions (Cubison & Jimenez, 2015) (Fig. B1 and Table B1) show that the peak intensities of HPMTF (p) and $IO_3^-CH_3COOH$ (p) in combination with our current level of mass resolution (~5000 m/$\Delta$m) were too low to be properly resolved (Sect. 2.2.2). This is in line with previous results from Wollesen de Jonge

et al. (2021), which showed that HPMTF was an important gaseous DMS oxidation product, but that is does not contribute directly in significant amounts to particulate mass. However, it is possible that HPMTF (g) is taken up by particles, especially when these are wetted (Vermeuel et al., 2020), where it likely would quickly undergo peroxide oxidation or nucleophilic substitution (Jernigan et al., 2022). Similar to earlier observations by Veres et al. (2020) and Wollesen de Jonge et al. (2021), the levels of HPMTF (g) during NASCENT were seen to decrease as a function of increased atmospheric water content. This

is further discussed in Sect. 3.3.

Further insights into the seasonality of the particle-phase DMS oxidation products can be obtained through comparison to the total particle number concentration (Fig. 3g) and concentrations of PM**x** (particulate matter < **x** μm) (Fig. 3h). The highest PM concentrations (Fig. 3h) were measured during the winter and spring (November–April), and the

lowest during the summer months (June–September). The peak in July, likely influenced by a biomass burning event, is an

exception. This pattern is in overall agreement to previous observations in Ny-Ålesund (Tunved et al., 2013). However, the

summer months are known for higher particle number concentrations due to new particle formation (Tunved et al., 2013),

where condensation of SA from DMS oxidation and formation of MSA are the main drivers (Xavier et al., 2022). Fig. 3g

clearly shows that the particle number concentration peak in 2020 coincided with the summer peak of particulate DMS

oxidation products, indicating that SA and MSA (but not HPMTF) were participating in the growth of the newly formed

particles. This means that the measured SA (p) in winter most likely was of anthropogenic origin and condensed onto pre-

existing accumulation mode particles in the atmosphere, whereas it in summer was produced locally via DMS oxidation and

at least partly via new particle formation.

### 3.2 Correlation between gas- and particle phase concentrations

As was discussed in the previous section, there appears to be a seasonal pattern of the DMS oxidation products, which is clearer

for MSA and SA than for HPMTF. As MSA and SA are known to contribute to particle mass (e.g. Lovejoy, Curtius and Froyd,

2004; Sipilä et al., 2010), a correlation between the gas- and particle phase can give indications on the direct relationship

between the two phases, which would be important for e.g. model simulations of particle growth. Previous studies from the

Arctic pack ice region in summer, farther away from DMS sources, have shown that MSA and SA do not exhibit any correlation

between the gas- and particle phase (Chang et al., 2011; Siegel et al., 2021), and similar results from Antarctica have been

reported (Davis et al., 1998; Read et al., 2008).

In Fig. 4 we show scatterplots for the two phases per "season", where January–April (JFMA, panel (a)) represent late

winter–early spring, May–July (MJJ, panel (b)) represent late spring–summer, and September–December (SOND, panel (c))

represent early fall–early winter.

As expected from Fig. 3, HPMTF appears to have no relationship between the phases in either season. The same is

true for SA (g) and (p) in JFMA and SOND. In MJJ a correlation can be noted between the gas- and particle phase for both

MSA and SA (Fig. 4b). To some extent, this correlation is visible also in JFMA for MSA (due to the increasing levels in April),

but not at all in SOND, after the bloom season. Furthermore, there appears to be a connection between the MSA and SA

correlations in MJJ, indicating that they have the same sources and are part of the same reaction processes. An orthogonal



linear regression analysis (non-weighted) (Cantrell, 2008) of the combined MSA and SA dataset (both gas- and particle-phase

signals logarithmized) in MJJ results in a an $R^2$ of 0.32, which indicates a weakly positive linear relationship. Due to the

insignificant levels of HPMTF (p), HPMTF lies outside of this connected relationship.

Besides seasonal variation, diurnal cycles of DMS oxidation products have previously been reported (Vermeuel et

al., 2020). However, no diurnal patterns were possible to see with our dataset, similarly to the study by Baccarini et al., 2021

in the Southern Ocean. Our hypothesis is that the large seasonal differences and fast changes around the equinoxes in Ny-

Ålesund affect the pattern in a way that makes diurnal cycles difficult to identify.

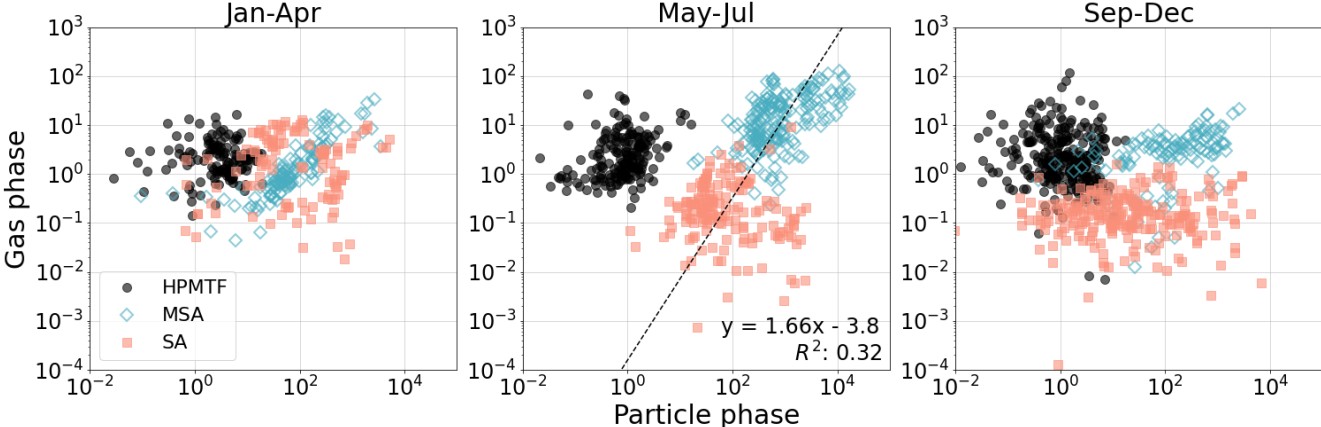

**Figure 4.** Relationship between gas- and particle phase MSA, SA and HPMTF per season: (a) January–April (JFMA), (b) May–July

(MJJ), (c) September–December (SOND). The black line in panel (b) represents the orthogonal linear regression between the combined

logarithmized MSA and SA datasets. The linear equation and correlation coefficient $R^2$ are shown in the lower right corner.

**3.3 Correlation of HPMTF (g) with visibility and relative humidity**

Previous studies have found gas-phase HPMTF to be readily taken up by clouds and hence suggested that wet scavenging is

an important atmospheric sink (e.g., Khan et al., 2021; Novak et al., 2021). Therefore, we investigated the measured levels of

HPMTF (g) as a function of visibility (vis, Fig. 5a) and RH (Fig. 5b) (averages per sampling period) during the summer months

(MJJ). Cloudy conditions are represented by vis < 1 km (the World Meteorological Organization's definition of fog; WMO,

2018) and cloud-free conditions by vis > 5 km. Vis = 1–5 km be viewed as a "transition state" between these conditions, here

referred to as "semi-cloudy". Further, a high atmospheric water content (referred to as "wet") is here defined as RH > 95%,




whereas a low water content (referred to as "dry") is set to < 75%. Hence, the range 75–95% ("semi-wet") refer to data points at neither high nor low RH.

In Fig. 5a, the average HPMTF (g) level is lower during cloudy and semi-cloudy compared to cloud-free conditions. However, these data are too unequally distributed (see the number of data points above each box-and-whisker plot) and skewed (especially > 5 km) to make a statistical analysis meaningful. The data related to RH in Fig. 5b are more equally distributed; however, they are skewed, as seen on the outliers towards higher HPMTF (g) levels. A statistical evaluation using the Wilcoxon rank-sum test for skewed data (two-tailed) of the three RH ranges, where the $p$-values are summarized in Table C1, shows that

the largest differences lie in the HPMTF (g) data between the wet case and the semi-wet/dry cases. These are statistically different at a 99% confidence level, whereas the data of the wet and semi-wet cases are statistically different at a 90% confidence level. The $p$-values of the analyses involving the wet case are, however, much smaller ($3.9\times10^{-18}$ and $3.0\times10^{-19}$) than between the semi-wet and dry cases (0.059), showing that HPMTF (g) is most effectively scavenged when RH > 95%.

Nevertheless, one must keep in mind that changes in RH could be sign of an air mass shift, which likely has an

influence on the atmospheric composition. Hence, it is difficult to conclude whether the RH or air mass origin is the main reason for the lower HPMTF (g) concentration at high RH. However, during the cloud events studied in Gramlich et al. (2022), HPMTF (g) levels were lower compared to right before and right after the cloudy period (Fig. 5c). These cloudy periods also correspond to the time with the highest RH values, hence, this demonstrates that cloudy and wet conditions do reduce HPMTF (g) levels in the atmosphere in Svalbard during summer. However, we did not detect HPMTF nor any of its possible

reaction products (e.g., $C_2H_4SO_2$, $C_2H_6SO_2$) (Jernigan et al., 2022) in significant amounts in any of the cloud residual samples (data not shown) (Gramlich et al., 2022). Therefore, we cannot make a conclusive statement about the fate of HPMTF in the particle phase with our dataset.



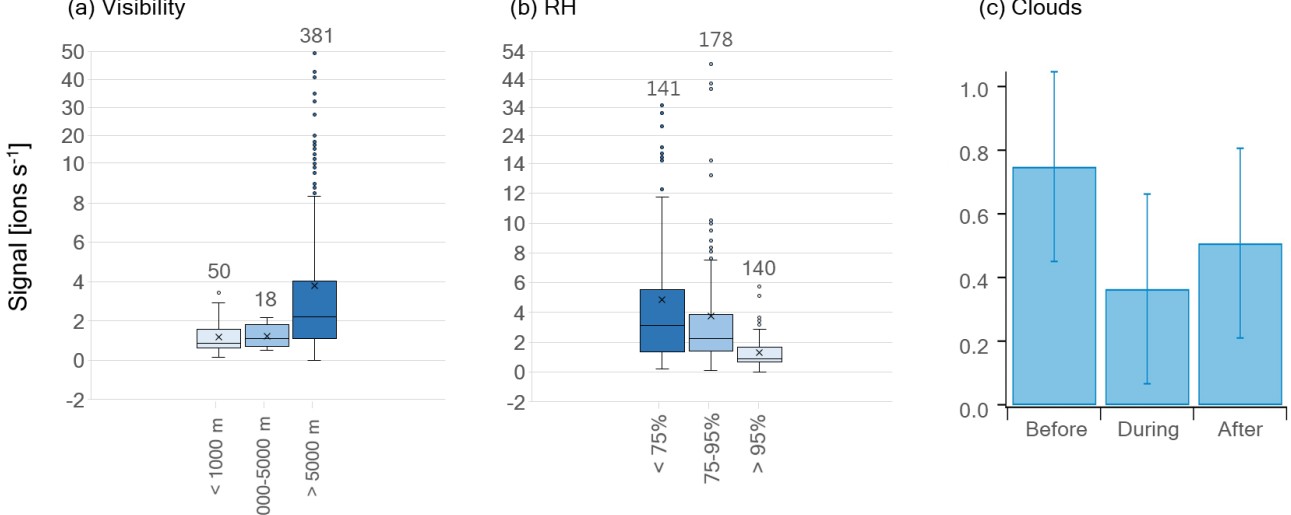

**Figure 5.** Box-and-whisker plots of HPMTF (g) levels during MMJ divided into (a) visibility ranges, (b) RH ranges (averages per sampling period). The horizontal lines inside the boxes correspond to the median and cross markers to the mean of each population. Quartiles are represented by the box borders, maximum and minimum values of the populations are represented by the whiskers, and outliers are shown as individual points outside the whisker ranges. The number of data points in each population is written above the plots. Panel (c) shows the average signal of HPMTF (g) before, during and after cloud events during May and June 2020. The error bars correspond to the standard deviation. The signals were normalized to the maximum HPMTF (g) signal of the individual cloud cases (combining the signals before, during and after) before computing the average and standard deviation.

## 3.4 Source regions of observed HPMTF

DMS is not produced close to Ny-Ålesund during wintertime due to the lack of sunlight, and the relatively high oberserved levels of presumed HPMTF (g) in January and high levels in October (Fig. 3e) are surprising. To investigate the possibilities of measuring high HPMTF concentrations during the dark and sunlit months, we analysed the HPMTF (g) content carried by



air masses to Ny-Ålesund in January, May and October using 5-day backward trajectories and satellite observations of marine cholorphyll-*a*, used as a proxy for DMS production (Fig. 6).

In May, the chlorophyll-*a* concentration was high in the coastal areas around Svalbard, and essentially all air masses arriving to Ny-Ålesund carried some HPMTF. The highest concentrations appear to originate from the Barents Sea, which is in line with previous results by Park et al. (2021).

In October, there were still some algal blooms at the Norwegian coast, south of Iceland and Greenland. However, the trajectories show that the air masses in October had spent the last 5 days before arriving to Ny-Ålesund within the Arctic region, far away from DMS-producing areas at lower latitudes. Several studies have shown that long-range transport of HPMTF is possible (Khan et al., 2021; Novak et al., 2021; Vermeuel et al., 2020); however, it seems unlikely as of the efficient scavenging of HPMTF by clouds (e.g., Khan et al., 2021; Novak et al., 2021). This is supported by the simulation case in the

study by Wollesen de Jonge et al. (2021), which could be said to represent our measurement conditions (Sect. 3.1.1) and included both in-cloud events and a rain event. Although is seems not likely, the possibility that HPMTF was transported to Svalbard in October cannot be ruled out.

In January there was almost no chlorophyll-*a* in the surface waters > 50° N, and most of the air masses originated from Europe and Sibiria. As the Arctic is experiencing polar night and low temperatures in January, and the trajectories indicate

that the air masses were arriving from terrestrial areas, another hypothesis is that these peaks should instead be attributed to $N_2O_5$, as has been observed previously in Alaska during wintertime (J. D. Ayers & Simpson, 2006). This hypothesis is partly supported by the fact that $N_2O_5$ was a better fit in the high-resolution mass spectrum than HPMTF during the second half of January (Fig. C3), although more directed measurements (such as in the study from Alaska) would be needed to verify this.





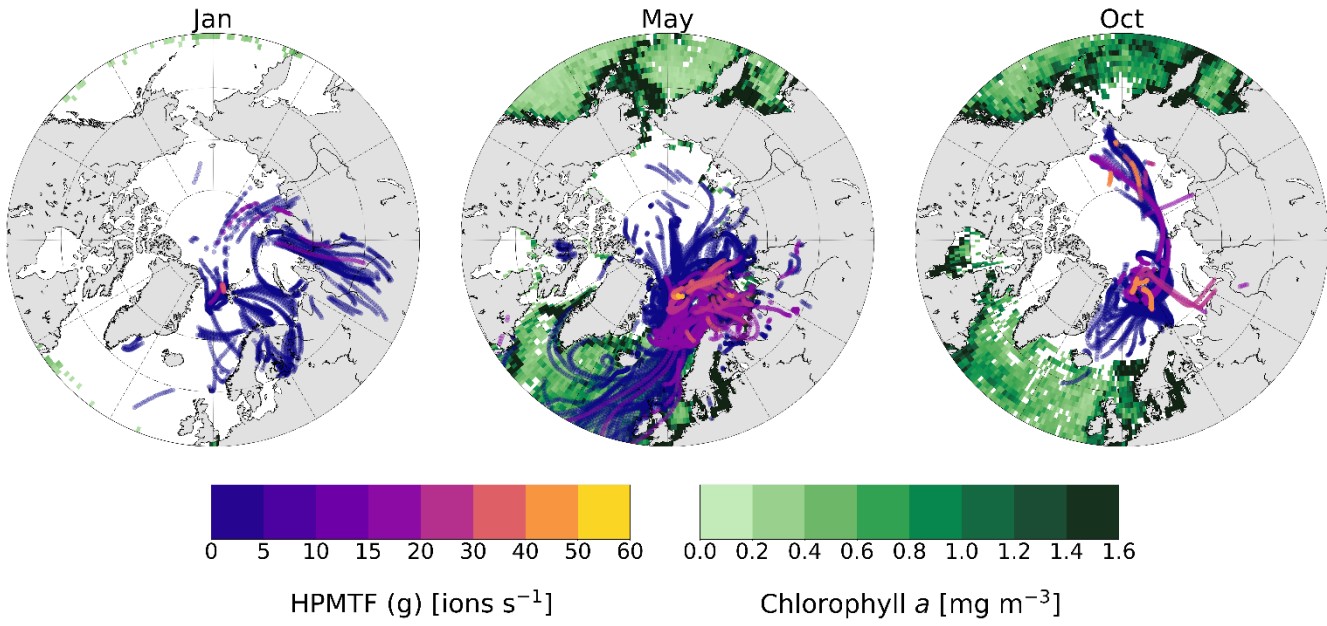

**Figure 6.** 5-day backwards trajectories arriving at the Zeppelin Observatory in (a) January, (b) May and (c) October, colored by measured HPMTF content in the gas phase [ions s⁻¹]. Only trajectory points corresponding to times below the mixed-layer height are shown. The green areas show marine chlorophyll *a* concentration (monthly mean), indicative of DMS production.

## 4 Summary and conclusions

We present a full year (2020) of gas- and particle phase observations of DMS oxidation products obtained with FIGAERO-CIMS at the Zeppelin Observatory, Svalbard. We focus especially on the newly discovered DMS oxidation product hydroperoxymethyl thioformate (HPMTF) (Veres et al., 2020), for which in situ observations are still limited. It has never been measured in Svalbard or attempted to be observed in atmospheric particulate matter.

Our results indicate that HPMTF is produced in large amounts in the vicinity of Svalbard during summer (April–September), and can be measured in October possibly as a result of transportation from lower latitudes. In summer, HPMTF follows the seasonal increase of MSA in the gas phase to some extent, but it could not be measured in the particle phase in significant amounts. This is in line with previous laboratory results by Wollesen de Jonge et al., (2021), stating that HPMTF did not contribute directly to particle growth despite high concentrations in the gas phase. Elevated levels in the gas phase

were also measured in January; however, due to the lack of DMS sources within a radius reasonable for long-range transport,

we speculate that this could instead be $N_2O_5$, which cannot be fully separated from HPMTF with the resolution of the mass

spectrometer deployed.

MSA follows a clear evolution pattern in both the gas- and particle phase, with an onset in April and fast decrease in

towards the end of Arctic summer (~September), similar to earlier findings in Ny-Ålesund (Jang et al., 2021). Sulfuric acid

(SA) was found in high concentrations during the winter/spring months (January–April) in both phases, assumed to be a result

of condensation onto accumulation mode particles from lower latitudes which are commonly found in the Arctic during the

spring haze period (e.g., Tunved et al., 2013). Particle-phase SA was also measured in summer when the total particle mass

concentrations were low but number concentrations high, indicating DMS oxidation and new particle formation as a source.

Due to the insignificant amounts of HPMTF in the particle phase, no correlation between the two phases could be

seen in any of the seasons. For MSA and SA, no correlation was visible in fall–winter–spring (~September–April), but a clear

relationship emerged during the sunlit months with DMS production (~May–July). The relationship between the phases of

MSA and SA also appeared to be connected, where MSA was more abundant in both phases during summer.

As has been seen in several previous studies (e.g., Veres et al., 2020; Vermeuel et al., 2020; Wollesen de Jonge et al.,

2021), we also noticed a decrease in HPMTF (g) levels during periods of cloud occurrence (visibility < 1 km) and high RH

(> 95%) in May–June. This shows that cloud removal is an efficient sink for HPMTF (g) also in Ny-Ålesund during summer.

With this study, we aimed to investigate the links between the relatively well-known seasonal patterns of MSA and

SA (Becagli et al., 2016; Jang et al., 2021; Leaitch et al., 2013) and the almost unknown HPMTF one. We concluded that there

is a relationship between gas-phase MSA and HPMTF in summer, indicative of an apparent contribution of the abstraction

pathway to DMS oxidation. SA and HPMTF have a low correlation, partly due to SA's preference for the particle phase and

HPMTF for the gas phase.

Future studies should focus on quantification of the measured levels of HPMTF in both gas- and particle phase. This

information could be used for modelling of DMS oxidation and would thereby increase quantitative knowledge about oxidation

pathways and sinks of HPMTF in the Arctic. For the particle-phase measurements in particular, it would be useful to investigate

the possibility of a better peak separation between HPMTF and mainly the acetic acid-iodate cluster $IO_3^-CH_3COOH$ in the

FIGAERO-CIMS. This can be achieved by e.g. using a mass spectrometer with a higher resolution or concurrent measurements

of $N_2O_5$ and acetic acid. The possibility for indirect contributions of HPMTF to particulate mass, such as an increase of sulfate

due to HPMTF oxidation in the aqueous phase as reported by Wollesen de Jonge et al. (2021), should also be investigated in

association to typical atmospheric conditions in the Arctic throughout the year.

## Data availability

The data of this study will be available on the Bolin Centre Database (https://bolin.su.se/data/). The meteorological data is

available on the EBAS data base (https://ebas-data.nilu.no).

## Author contribution

CM, PZ and RK were responsible for funding and conceptualization. YG, KS, SH and CM performed the FIGAERO-CIMS

measurements. YG, KS and SH wrote code for FIGAERO-CIMS analysis. RK, PZ and GF operated the GCVI and provided

the GCVI data. GF provided the DMPS data. KS and YG analyzed and visualized the FIGAERO-CIMS data, and visualized

the GCVI and DMPS data. KS visualized the trajectories, and wrote the manuscript with input from CM. CM was responsible

for supervision. All co-authors have read and commented on the manuscript.

## Competing interests

Some of the authors are co-editors at ACP.

## Acknowledgements


We are grateful for the financial support from the Knut and Alice Wallenberg (KAW) foundation (WAF project

CLOUDFORM, grant no. 2017.0165 and ACAS project # 2016.0024) and the European Union's Horizon 2020 research and

innovation programme under grant agreement No 821205 (FORCeS). The work was also financially supported by Swedish

Environmental Protection Agency (Naturvårdsverket) and by the funding agency FORMAS (IWCAA project # 2016-01427).



This work was supported by the Swedish Research Council (Vetenskapsrådet starting grant, project number 2018-05045 and

project number 2016-05100). We thank Peter Tunved for calculating the HYSPLIT trajectories. We are further grateful to the

Norwegian Institute for Air Research (NILU) for providing zero air, and especially Wenche Aas, Anne Hjellbrekke, and Ove

Hermansen for providing meteorological data. We owe great thanks to Helge Tore Markussen and Vera Sklet and the

technicians at the Norwegian Polar Institute (NPI) in Ny-Ålesund and ACES, especially Christer Sørem, Christelle Guesnon,

Håkon Jonsson Ruud, Svein-Torgar Oland Paulsen, Filip Heitmann, Ondrej Tesar and Tabea Hennig for all their support.



**Appendix A**

Details on the FIGAERO-CIMS measurement cycles, referring to the text in Sect. 2.2.


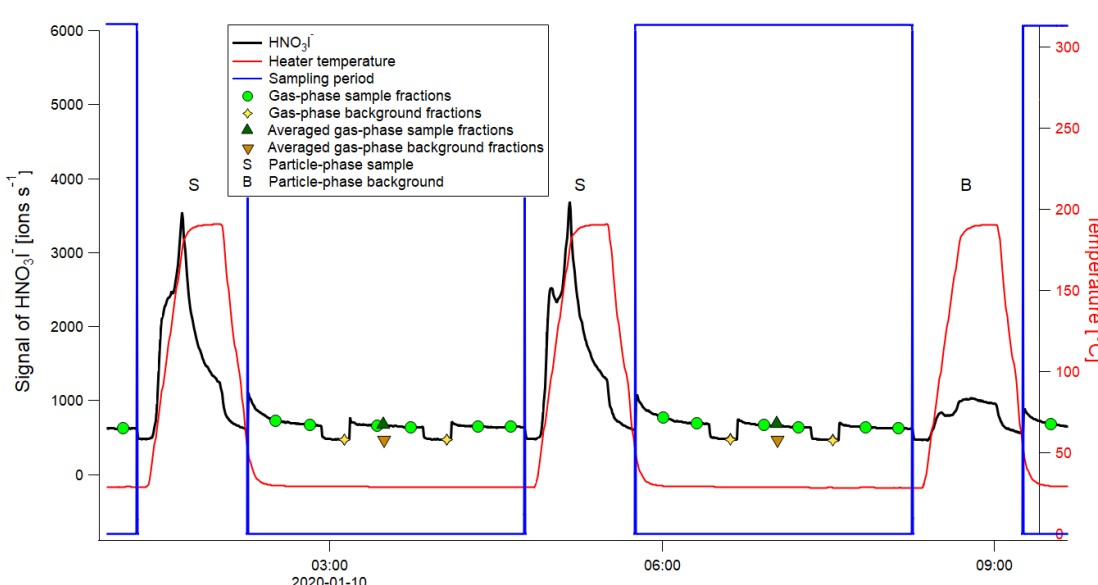

**Figure A1.** Example of a FIGAERO-CIMS sampling cycle, using the timeseries of the nitric acid iodide adduct ($HNO_3I^-$), shown in black. The letter S represents a particle-phase sample and B a particle-phase blank. The blue boxes define the times of a sampling cycle, when gas-phase measurements are done while particles are deposited on a filter. The green circle markers show the gas-phase sample segments (normally 3 segments) and the yellow diamond markers the gas-phase background segments (normally 2) identified for each sampling period.
The dark-green triangles represent the averaged gas-phase sample segments and the orange triangles the averaged gas-phase background segments. The solid red line shows the heater temperature, i.e. the temperature of the $N_2$ flow passing through the filter.

**Table A1.** Scheme of the FIGAERO-CIMS measurement cycle during NASCENT. Every third such cycle was particle blank with an additional particle filter upstream of the particle sample filter. BG stands for background measurement.

| | | 150 min | | | 60 min |
|---|---|---|---|---|---|
| **Particle phase:** | | Particle deposition | | | Particle desorption |
| **Gas phase:** | Sample | BG | Sample | BG | Sample | |
| | 40 min | 15 min | 40 min | 15 min | 40 min | |





## Appendix B

Peak separation of HPMTF and its neighbor peaks.


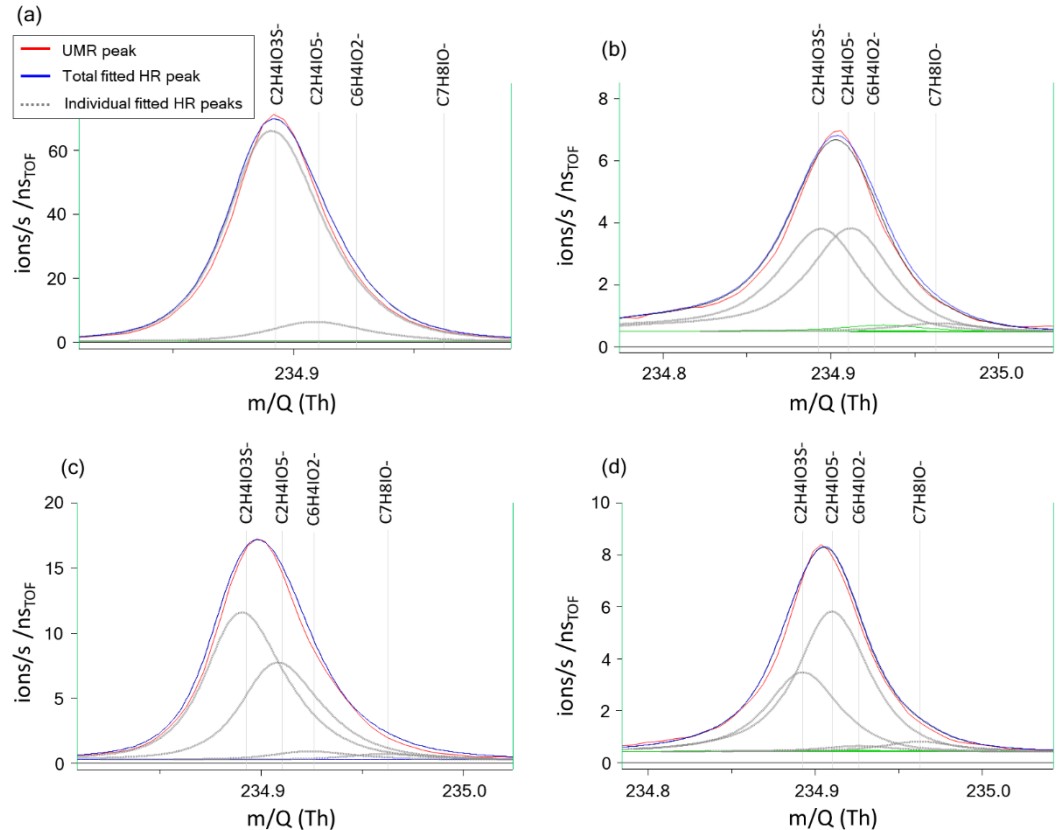

**Figure B1.** Four example cases of the separation of the two peaks HPMTF (in figure: C2H4IO3S-) and the acetic acid/iodate cluster $IO_3^-$·$CH_3COOH$ (in figure: C2H4IO5-) (Veres et al., 2020), as analyzed by FIGAERO-CIMS: (a) HPMTF as single ion (in practice a high HPMTF/$IO_3^-$·$CH_3COOH$ ratio), (b) HPMTF as the larger (parent) peak and $IO_3^-$·$CH_3COOH$ as the smaller (child) peak, (c) equal intensities of HPMTF and $IO_3^-$·$CH_3COOH$, and (d) $IO_3^-$·$CH_3COOH$ as the parent peak and HPMTF as the child peak. The peak separation statistics are shown in Table B1. UMR = Unit Mass Resolution, HR = High Resolution.




**Table B1.** Separation statistics of the two peaks HPMTF and the acetic acid/iodate cluster $IO_3^-CH_3COOH$ (Cubison & Jimenez, 2015). The first column corresponds to the example cases shown in Fig. B1 (a)-(d). The parameter $\chi$ shows the degree of separation of the two peaks, whereas $\sigma_B$ shows the statistical precision of the peak intensities. When in bold font, $\sigma_B$ corresponds to the precision of the HPMTF peak as parent/child. In case (c), HPMTF can be viewed as either the parent or child, and the precision is therefore in the range 3.2-5.4%.

| Peak type | Fitted intensity [ions] | HPMTF % of UMR intensity | Ratio HPMTF/ C2H4IO5 | Peak separation ($\chi$) | $\sigma_B$ (parent) [%][*] | $\sigma_B$ (child) [%][*] | Regime[**] |
|---|---|---|---|---|---|---|---|
| Single ion (a) | 212 | 91 | 11 | 0.7 | **51** | 2.9 | Counting-error |
| Parent (b) | 37.6 | 57 | 1.5 | 0.8 | **6.3** | 2.5 | Counting-error |
| Equal (c) | 14.8 | 48 | 1.0 | 0.6 | **5.4** | **3.2** | Counting-error (parent) Over-lapping counting error (child) |
| Child (d) | 11.2 | 34 | 0.56 | 0.7 | 8.6 | **2.9** | Overlapping counting-error |

[*] Calculated from eq. (3) and (4) in Cubison & Jimenez (2015), respectively.

[**] Estimate based on Fig. 5 in Cubison & Jimenez (2015).





## Appendix C

Additional field measurements and calculations.


**Figure C1.** Timeseries of (a) gas-phase MSA and HPMTF, (b) gas-phase inorganic SA and $H_2O_7S_2$, (c) gas-phase inorganic $SO_2$, $HSO_3$ and $SO_3$, (d) particle-phase sulfuric acid, MSA and HPMTF (note the different axis scales for the compounds).




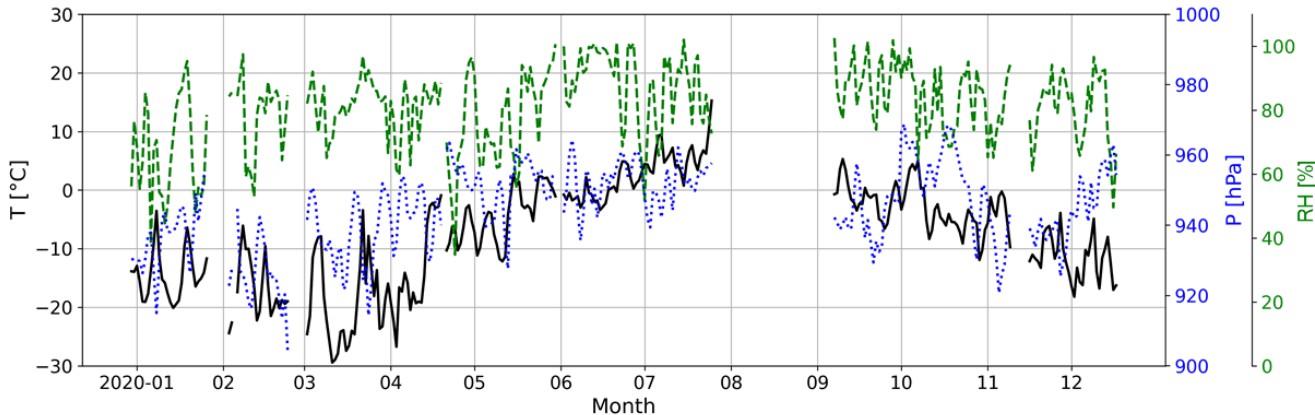


**Figure C2**. Atmospheric temperature (T, black solid line), pressure (P, blue dotted line) and relative humidity (RH, green dashed line) at the Zeppelin Observatory during NASCENT.

**Table C1.** Matrix showing the *p*-values of the statistical analysis (two-sided Wilcoxon rank-sum test) between HPMTF (g) levels at different RH during May-June (MJJ).

| RH | 75-95% | > 95% |
|---|---|---|
| < 75% | 0.059 | $3.9 \times 10^{-18}$ |
| 75-95% | - | $3.0 \times 10^{-19}$ |

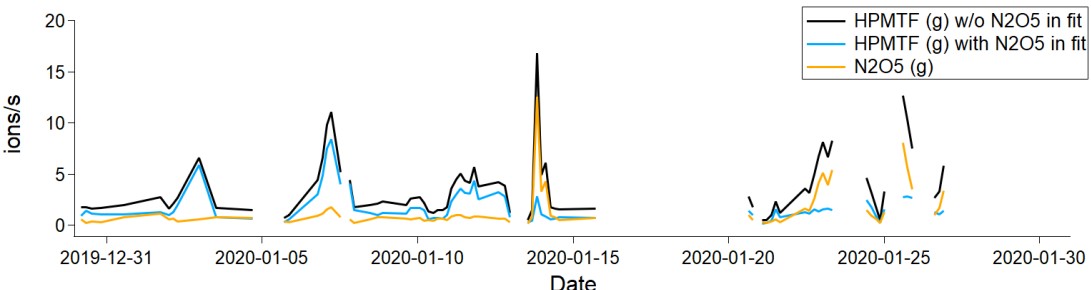

**Figure C3.** Average (per sampling period) gas-phase signal of HPMTF (g) and $N_2O_5$ (g) in January 2020. The blue and black lines show the HPMTF signal when $N_2O_5$ was and was not included in the peak fit, respectively. The orange line shows the $N_2O_5$ signal, in comparison to the blue HPMTF signal.



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
