# Peer review of "Arctic observations of Hydroperoxymethyl Thioformate (HPMTF) – seasonal behavior and relationship to other oxidation products from Dimethyl Sulfide at the Zeppelin Observatory, Svalbard"

_EGUsphere, 2023_

## Author Response (AR1)

**RC1**:

**Siegel et al. presented gas-phase and particle-phase measurements of SA, MSA and HPMTF during the full year of 2020 at the Zeppelin Observatory, Ny-Ålesund, Svalbard. They report high gas-phase concentrations of HPMTF between April and September when DMS emissions are high, but observe no significant concentration of HPMTF in the particle-phase. The paper is well written and provides an important insight into the role of HPMTF in the atmosphere. I recommend that the paper be published after addressing the comments bellow.**

The authors thank the reviewer for the positive assessment of our manuscript and the time invested to review it. The authors comment on each of the reviewer's comment (bold) individually. The lines given in the answers refer to the revised document. Figure numbers refer to Figures in the manuscript and Figure letters refer to Figures only provided for the review process.

**Specific comments:**

**Page 1 - line 2: "methanesulfonic acid (MSA) and sulfuric acid (SA) are well-known for participating in the formation and growth of atmospheric aerosol particles". While it is well known that SA contributes to new particle formation, I would not say that MSA is well-known to do so. Recent studies have found it plausible that MSA contributes to NPF, but it has not been definitely established.**

The authors thank the reviewer for pointing this out. The authors changed L14-17 to "DMS undergoes oxidation in the atmosphere to form a range of oxidation products, out of which sulfuric acid (SA) is well-known for participating in the formation and growth of atmospheric aerosol particles, and the same is also presumed for methanesulfonic acid (MSA)."

**Page 2 - line 29: "Up to 42% of global natural sulfur emissions can be traced back to DMS". This is a low estimate, and other studies have reported that DMS may comprise more than 50% of the global natural sulfur emission. Therefore, I would not use the phrasing, 'up tp 42%'.**

The authors rephrased L32-34 to "The global natural DMS emissions vary largely between the southern and the northern hemisphere. On a global average, around 42% of the natural sulfur emissions can be traced back to DMS (Simó, 2001), which is equal to at least 50% of the total amount from anthropogenic sources (Simó, 2001; Klimont et al., 2013)".

**Figure 1: I find the schematic for SO2 and SO3 production a bit confusing. While SO2 produces SO3 which in turn produces SA, SO3 is also formed directly from CH3SO3 (which is the dominant pathway leading to SO3 and thus SA production from DMS). Consider making an arrow that branches into both SO2 and SO3 production to indicate that they are produced from two separate pathways.**

The authors modified Fig. 1 accordingly, see below. We also include methanesulfonate (CH₃SO₃) in the schematic, as mentioned by the reviewer, and updated L51-56 to "Main products in the abstraction pathway are the inorganic compound sulfuric acid (SA, H₂SO₄) (via sulfur dioxide, SO₂ or methanesulfonate (CH3SOOO˙), and sulfur trioxide, SO₃), and the organic compound methanesulfonic acid (MSA, CH₃SO₃H). The first stable product in the addition pathway is dimethyl sulfoxide (DMSO, CH₃SOCH₃), followed by methanesulfinic acid (MSIA, CH₃S(O)OH) (Barnes et al., 2006). MSIA can either undergo reactive uptake to the particle phase or oxidize further via methanesulfonate to MSA and SA, although the abstraction pathway is normally considered more important for the production of these two species.".

[Figure]

Figure 1. Simplified oxidation scheme of dimethyl sulfide (DMS) in the atmosphere. DMS is produced by microbiological activity in the ocean and emitted to the atmosphere, where it is oxidized through two main routes: 1) *addition* of a radical to produce dimethyl sulfoxide (DMSO) and methanesulfinic acid (MSIA), and further via methanesulfonate (CH₃SO₃) to methanesulfonic acid (MSA) and/or sulfuric acid (SA) either via sulfur dioxide (SO₂) and sulfur trioxide (SO₃) or via methanesulfonate and SO₃; 2) *abstraction* of a hydrogen (H) atom to produce MSA, hydroperoxymethyl thioformate (HPMTF; marked with a box) and/or SA via SO₂ and SO₃. The figure was created using information from (Barnes et al., 2006 and R. Wu et al., 2015). The addition pathway is shown in orange arrows, the abstraction pathway

in green arrows, and DMS oxidation products that are part of both pathways are indicated with black arrows.

**Page 15 - line 296: "the summer months are known for higher particle number concentrations due to new particle formation (Tunved et al., 2013), where condensation of SA from DMS oxidation and formation of MSA are the main drivers". Rephrase this statement. It is not the condensation of SA and MSA that drives the increase in PN. It is the new particle formation from SA (and maybe also MSA) that causes an increase in PN.**

The authors thank the reviewer for indicating this and rephrased L311-313 accordingly: "However, the summer months are known for higher particle number concentrations due to new particle formation (Tunved et al., 2013), driven by SA (g) with subsequent growth by condensation of SA (g) and MSA (g) (Beck et al., 2021; Xavier et al., 2022)."

**Figure 4: Why would you show a combined R2 value for SA and MSA, and not just report R2 for both species?**

We show a combined R2 value for SA and MSA to visualize that they can have the same sources and are part of the same reaction processes in the summer months MJJ. Presenting separate R2 values for MSA and SA would only indicate the relation between the gas- and the particle phase for the two compounds individually, which is not what the authors intended to emphasize. This has been made clearer in the new version of the manuscript (L330-337). To better visualize the correlation of the combined MSA and SA data, the authors made the figure below, where no distinction is made between MSA and SA points. A linear relationship then appears more clearly. Please note that the figure below is for the reviewer's information only.

[Figure]

Figure a: Relationship between the gas- and particle phase of MSA, SA, HPMTF per season, where SA and MSA have the same color to show the connection between the combined MSA and SA correlation.

**Technical comments:**

**Figure 2: Consider using (a), (b), (c), (d) instead of upper left, upper right, etc.**

Changed as suggested, see updated Fig. 2 below.

[Figure]

Figure 2. (a) Map of Svalbard, where Ny-Ålesund is marked with a red circle. (b) Mt. Zeppelin with the Zeppelin Observatory in relation to Ny-Ålesund (maps generated using Python's Matplotlib Basemap toolkit and ©OpenStreetMap contributors 2023, distributed under the Open Data Commons Open Database License (ODbL) v1.0). (c) View of Mt. Zeppelin and the observatory from Ny-Ålesund. (d) View of Ny-Ålesund and Kongsfjorden from the Zeppelin Observatory (photos taken in September 2021).

**Page 10 - line 218: "to be able to produce DMS". Write, "in order to produce DMS".**

Changed as suggested.

**Figure 4: No need to have the same y-axis ticks all three plots. Just keep the ones on the left plot.**

Changed as suggested, see below.

[Figure]

Figure 3. Relationship between gas- and particle-phase MSA, SA and HPMTF per season: (a) January–April (JFMA), (b) May–July (MJJ), (c) September–December (SOND). The black line in panel (b) represents the orthogonal linear regression between the combined logarithmized MSA and SA datasets. The linear equation and correlation coefficient $R^2$ are shown in the lower right corner.

**Figure 4: What is the unit for the gas-phase and particle-phase measurements?**

The authors thank the reviewer for indicating that the units are missing in Fig. 4. The unit of the gas phase is ions per second, and the unit of the particle phase is ions per liter. The units have been added to the x- and y-axis label.

**Page 19 - line 381: "Although it seems not likely". Write, "Although it seems unlikely".**

Changed as suggested.

**Page 21 - Line 421: "and the almost unknown HPMTF one". Write, "and the almost unknown one of HPMTF".**

Changed as suggested.

**RC2**:

**The manuscript "Arctic observations of Hydroperoxymethyl Thioformate (HPMTF) - seasonal behaviour and relationship to other oxidation products from Dimethyl Sulfide at the Zeppelin Observatory, Svalbard" focuses on a 1-year dataset of gas- and particle phase observations of HPMTF, MSA, SA, SO2, SO3, HSO3, H2O7S2, particle number concentration and particle mass in the Arctic. The special focus of the manuscript is on HPMTF, assessing its measurability in the particle phase and understanding its sources. I believe that the manuscript is interesting to the scientific community and thus I recommend the paper for publication after the following comments have been addressed:**

The authors thank the reviewer for the positive assessment of our work and the time spend to read and provide comments on the manuscript to enhance its quality. We respond to each of the reviewer's comments (bold) individually (normal font). The line numbers in the responses refer to the revised manuscript. Figure numbers refer to Figures in the manuscript and Figure letters refer to Figures only provided for the reviewer.

**General comments:**

**Unique data: The data from this campaign is split into several papers. I find it a bit unclear which data is only being presented in this paper and which one has already been presented before. I suggest to more clearly state which data is new in this paper. This could fit for example at the end of the introduction section or the methods section.**

The authors thank the reviewer for this comment. We added a statement about the types of datasets obtained from the FIGAERO-CIMS to the beginning of the methods section, L108-110: "Our FIGAERO-CIMS setup at the Zeppelin Observatory yielded two types of datasets: one on the aerosol particle composition that acted as CCN or INPs (also termed cloud residuals, reported in Gramlich et al. (2022)), and one on the ambient aerosol particle and gas phase composition, which is reported in this study."

**Visibility measurement: The paper would profit from some more information on the visibility measurement. How does the technique work? At which elevation was the visibility assessed? Which conditions define fog?**

The authors added information about the visibility in the methods section, L128-132: "During cloudy conditions, defined as when the visibility (visibility sensor from Belfort, Model 6400, at approx. 480 m a.s.l.) at the observatory was < 1000 m (World Meteorological Organization definition of fog (WMO, 2018)), the three-way valve switched to sampling cloud particles (> 6–7 μm in diameter) through a ground based counterflow virtual impactor (GCVI; Brechtel Manufacturing Inc., USA, Model 1205) inlet (Karlsson et al., 2021)." The authors removed the statement about the definition of fog in the results section from L349-351, which is now "Therefore, we investigated the measured levels of HPMTF (g) as a function of visibility (vis, Fig. 5a) and RH (Fig. 5b) (averages per sampling period) during the

summer months (MJJ). Cloudy conditions are represented by vis < 1 km and cloud-free conditions by vis > 5 km".

**Particle number concentration: what is the cut-off of the CPC?**

The lower cut off size of the CPC (TSI Model 3772) is 5 nm, and the upper size cut is defined by the whole air inlet, which is approx. 40 µm. The authors added this information in L205-207: "A condensation particle counter (CPC; TSI Inc., USA, Model 3772) behind the whole-air inlet was used to measure the total particle number concentration (1 Hz data averaged to 1 min time resolution, lower cut-off size 5 nm, upper cut-off size approx. 40 µm)."

**P4, line 77: I think it could be good to mention that when you compare to Wollesen de Jonge et al. (2021) it is not experimental results of HPMTF you compare to but model results. Other things like MSA and DMS were measured in chamber experiments but not HPMTF itself.**

The authors thank the reviewer for pointing this out. We modified L83-85 accordingly: "Wollesen de Jonge et al. (2021) showed in a model study based on chamber experiments that HPMTF accumulates in the gas phase during cloud-free conditions, but does not significantly partition to the particle phase and is unlikely to contribute to NPF."

**P6, line 115: You state that the cut-off of the inlet is approximately 40 µm. Could you please specify how you assessed this? For example, did you use the particle loss correction program?**

The authors did not assess this themselves. The inlet follows the guidelines from the World Calibration Centre for Aerosol Physics (WCCAP) at the Leibniz Institute for Tropospheric Research, Germany, and is technically identical to the inlet described by Weingartner et al. (1999), which has the specification to sample particles up to 40 µm. The authors modified L123-127 to "The aerosol particles were sampled through a gently heated whole-air inlet, which follows the guidelines of the World Calibration Centre for Aerosol Physics (WCCAP) at the Leipniz Institute for Tropospheric Research, Germany (Wiedensohler et al., 2013). The inlet samples all aerosol and cloud particles smaller than approx. 40 µm at wind speeds up to 20 m s$^{-1}$ (Weingartner et al., 1999), and is located at ~480 m a.s.l..".

**Chapter 2.4.: you mention the "model mixed layer height" – could you please specify a little more which altitudes we are talking about? What was on average the mixed layer height?**

The average monthly mixed-layer height is around 260 m, 380 m, and 390 m in Jan, May and Oct, respectively. The authors added L216-217 "The average monthly mixed-layer was around 260 m, 380 m, and 390 m in January, May and October, respectively." In the figure below, we show the histograms of the monthly mixed layer heights for the reviewer.

[Figure]

Figure b: Histograms of the averaged mixed-layer heights along the trajectories in January, May and October, respectively. The y-axis shows the absolute number of the occurrence of the respective bin.

**P15, line 316 – Fig. 4b: I have a hard time seeing a correlation of SA between gas and particle phase in this plot. I can see that it falls on the regression line of MSA but to me it looks more like a cloud of particles (similar to the one seen in Fig. 4a, where you say that there is no clear correlation). How did you assess that there is a correlation in 4b and not in 4a?**

Compare also response to comment on Figure 4 by reviewer 1. The reviewer is right that there is no clear correlation of SA between the gas- and particle phase. With Figure 4 we want to show that combining the MSA and SA gas- and particle-phase signal together yields a correlation for the summer months MJJ but not for the other seasons, meaning that in MJJ, MSA and SA may both come from the same source region. The authors clarified this in L330-337: "As expected from Fig. 3, HPMTF appears to have no relationship between the phases in either season. The same is true for SA (g) and (p). To some extent, a correlation between the gas and particle phase is visible for MSA in JFMA and MJJ (due to the increasing levels in April and high production during summer), but not at all in SOND (after the bloom season). Although no correlation is notable for MSA and SA individually in MJJ, a different pattern appears when combining the datasets. An orthogonal linear regression analysis (non-weighted) (Cantrell, 2008) of the combined MSA and SA data (both gas- and particle-phase signals logarithmized) in MJJ results in a an $R^2$ of 0.32 (Fig. 4b), which indicates a weakly positive linear relationship, and that

MSA and SA have the same sources and are part of the same reaction process during these months. Due to the insignificant levels of HPMTF (p), HPMTF lies outside of this connected relationship."

**Specific comments:**

**Chapter 3.1.1: for clarity I would propose to either omit the specification of "(g)" or write it consistently in all occasions in the paragraph to avoid misunderstandings. You could omit it as it is clear from the title. Similarly in chapter 3.1.2 I would either specify it in all cases or in none.**

The authors decided to write (g) and (p) consistently down in all occasions.

**P10, line 222: delete "and" after MSA.**

Changed as suggested.

**P11, line 236: please reformulate "develop similarly to MSA to some degree" to make it more precise**

Changed to "The levels of HPMTF (g) (Fig. 3c) develop similarly to MSA (g) during some parts of the year, with low levels in March, an increase in April and a peak in May."

**P11, line 237: write "and peaking in May" instead of "to a peak"**

Changed to "and a peak in May".

**P12, line 255: what is meant here by "relative temporal evolution"?**

The authors clarified L270-272 as following: "However, the discrepancies that exist in the monthly absolute signal between measured MSA (g) and HPMTF (g) in summer could also be a sign that HPMTF is not produced as efficiently as MSA close to Svalbard due to e.g. low occurrence of OH radicals and/or meteorological factors."

**P14, line 275: replace "particle- to the gas phase" by "particle and gas phase"**

Changed as suggested.

**P15, line 300: I am a bit confused about the sentence: "This means that the measured SA(p)…". You talk about SA(p) and condensation onto particles. Don't you mean that SA(g) condensed on the particles to become SA(p)?**

The authors thank the reviewer for pointing out this misconception. The authors changed L316-318 to "This means that the measured SA (p) in winter most likely was of anthropogenic origin where SA (g) condensed onto pre-existing accumulation mode particles in the atmosphere, whereas in summer it was produced locally via DMS oxidation and subsequent condensation and at least partly via new particle formation."

**P17, line 349: insert an "a" between "be" and "sign"**

Changed as suggested.

**P21, line 407: delete the "in" after "in April and fast decrease.."**

Changed as suggested.

**Author contributions: I am confused why it says something about DMPS data, is the CPC data meant? I couldn't find particle size distributions anywhere?**

The authors thank the reviewer for pointing this out. CPC data was meant here. The authors clarified this accordingly, L454: "PZ provided the CPC and the FIDAS data. KS and YG analyzed and visualized the FIGAERO-CIMS data, and visualized the GCVI and CPC data."

**Additional edits by the authors:**

L133: Removed "first" from "the first results of this analysis are reported in Gramlich et al. (2022)."

Change of the email address and the affiliation of the corresponding author.

Acknowledgements: The authors also thank Chris Lunder and Ian (Gang) Chen for providing the ACSM data.

**References**

Barnes, I., Hjorth, J., and Mihalopoulos, N.: Dimethyl Sulfide and Dimethyl Sulfoxide and Their Oxidation in the Atmosphere, Chem. Rev., 106, 940–975, https://doi.org/10.1021/cr020529+, 2006.

Beck, L. J., Sarnela, N., Junninen, H., Hoppe, C. J. M., Garmash, O., Bianchi, F., Riva, M., Rose, C., Peräkylä, O., Wimmer, D., Kausiala, O., Jokinen, T., Ahonen, L., Mikkilä, J., Hakala, J., He, X., Kontkanen, J., Wolf, K. K. E., Cappelletti, D., Mazzola, M., Traversi, R., Petroselli, C., Viola, A. P., Vitale, V., Lange, R., Massling, A., Nøjgaard, J. K., Krejci, R., Karlsson, L., Zieger, P., Jang, S., Lee, K., Vakkari, V., Lampilahti, J., Thakur, R. C., Leino, K., Kangasluoma, J., Duplissy, E., Siivola, E., Marbouti, M., Tham, Y. J., Saiz-Lopez, A., Petäjä, T., Ehn, M., Worsnop, D. R., Skov, H., Kulmala, M., Kerminen, V., and Sipilä, M.: Differing Mechanisms of New Particle Formation at Two Arctic Sites, Geophys Res Lett, 48, https://doi.org/10.1029/2020GL091334, 2021.

Cantrell, C. A.: Technical Note: Review of methods for linear least-squares fitting of data and application to atmospheric chemistry problems, Atmos. Chem. Phys., 8, 5477–5487, https://doi.org/10.5194/acp-8-5477-2008, 2008.

Karlsson, L., Krejci, R., Koike, M., Ebell, K., and Zieger, P.: A long-term study of cloud residuals from low-level Arctic clouds, Atmos. Chem. Phys., 21, 8933–8959, https://doi.org/10.5194/acp-21-8933-2021, 2021.

Klimont, Z., Smith, S. J., and Cofala, J.: The last decade of global anthropogenic sulfur dioxide: 2000–2011 emissions, Environ. Res. Lett., 8, 014003, https://doi.org/10.1088/1748-9326/8/1/014003, 2013.

Simó, R.: Production of atmospheric sulfur by oceanic plankton: biogeochemical, ecological and evolutionary links, Trends in Ecology & Evolution, 16, 287–294, https://doi.org/10.1016/S0169-5347(01)02152-8, 2001.

Tunved, P., Ström, J., and Krejci, R.: Arctic aerosol life cycle: linking aerosol size distributions observed between 2000 and 2010 with air mass transport and precipitation at Zeppelin station, Ny-Ålesund, Svalbard, Atmos. Chem. Phys., 13, 3643–3660, https://doi.org/10.5194/acp-13-3643-2013, 2013.

Weingartner, E., Nyeki, S., and Baltensperger, U.: Seasonal and diurnal variation of aerosol size distributions ($10<D<750$ nm) at a high-alpine site (Jungfraujoch 3580 m asl), J. Geophys. Res., 104, 26809–26820, https://doi.org/10.1029/1999JD900170, 1999.

Wiedensohler, A., Birmili, W., Putaud, J.-P., and Ogren, J.: Recommendations for Aerosol Sampling, in: Aerosol Science, edited by: Colbeck, I. and Lazaridis, M., John Wiley & Sons, Ltd, Chichester, UK, 45–59, https://doi.org/10.1002/9781118682555.ch3, 2013.

WMO: Measurement of Meteorological Variables, 2018.

Wollesen de Jonge, R., Elm, J., Rosati, B., Christiansen, S., Hyttinen, N., Lüdemann, D., Bilde, M., and Roldin, P.: Secondary aerosol formation from dimethyl sulfide – improved mechanistic understanding based on smog chamber experiments and modelling, Atmos. Chem. Phys., 21, 9955–9976, https://doi.org/10.5194/acp-21-9955-2021, 2021.

Wu, R., Wang, S., and Wang, L.: New Mechanism for the Atmospheric Oxidation of Dimethyl Sulfide. The Importance of Intramolecular Hydrogen Shift in a $CH_3SCH_2OO$ Radical, J. Phys. Chem. A, 119, 112–117, https://doi.org/10.1021/jp511616j, 2015.

Xavier, C., Baykara, M., Wollesen de Jonge, R., Altstädter, B., Clusius, P., Vakkari, V., Thakur, R., Beck, L., Becagli, S., Severi, M., Traversi, R., Krejci, R., Tunved, P., Mazzola, M., Wehner, B.,

Sipilä, M., Kulmala, M., Boy, M., and Roldin, P.: Secondary aerosol formation in marine Arctic environments: a model measurement comparison at Ny-Ålesund, Atmos. Chem. Phys., 22, 10023–10043, https://doi.org/10.5194/acp-22-10023-2022, 2022.